# Dynamic Semantic Routing for Multimodal Sentiment Analysis

## Abstract

Multimodal sentiment analysis (MSA) aims to understand human emotions by integrating heterogeneous signals such as language, vision, and acoustic modalities. However, multimodal data often suffer from internal semantic entanglement, ambiguous cues, and inconsistent modality contributions, which limit the effectiveness of unified representations. To address these challenges, we propose a Dynamic Semantic Routing Framework (DSRF) for the MSA task. Specifically, we present a hierarchical semantic factorization module, which disentangles each modality into four functionally independent representations: primary emotion, contextual cue, ambiguity, and noise, enabling fine-grained semantic modeling. Moreover, we introduce a semantic dynamic routing interaction mechanism, which dynamically routes and aggregates the semantic factors through a capsule-inspired interaction process to reconstruct modality representations with high-order compositionality. Finally, we design an uncertainty-aware semantic fusion strategy that estimates the reliability of each semantic factor and adaptively integrates them across modalities for robust sentiment prediction under modality inconsistency. Extensive experiments on four benchmark datasets demonstrate that our framework achieves state-of-the-art performance.

## 1 Introduction

Multimodal Sentiment Analysis (MSA) has emerged as a crucial subfield of affective computing, enabling machines to understand human emotional states by leveraging diverse signals such as spoken language, facial expressions, and vocal prosody Liu et al. (2022a); Akbari et al. (2021); Yang et al. (2022b); Li et al. (2023a); Yang et al. (2024). These heterogeneous modalities are inherently complementary: language conveys explicit sentiment cues, while audio and visual signals offer nuanced paralinguistic information Wang et al. (2019); Shad Akhtar et al. (2019); Sun et al. (2020). The integration of these signals allows for a more holistic understanding of emotion, which is especially valuable in applications such as empathetic dialogue systems, social media analysis, and intelligent healthcare Zhai et al. (2022); Zhang et al. (2021).

With the advancement of deep learning, many MSA methods have explored fusion mechanisms to align and combine features across modalities. Early models employed string-based concatenation and tensor fusion Ngiam et al. (2011); Zadeh et al. (2018b), while more recent work has adopted attention-based interaction frameworks Amjad & Geiger (2019); Tsai et al. (2019), allowing for fine-grained modeling of inter-modal dependencies. For instance, MICA Liang et al. (2021) captures cross-modal dynamics via attention, and TAILOR Zhang et al. (2022) gradually fuses features through a hierarchical transformer. These models assume that the unified representation can sufficiently encode emotional semantics by combining all available information.

Nevertheless, such unified modeling approaches often overlook the internal semantic complexity of each modality. Sentiment expressions are not monolithic—each modality may simultaneously carry core emotion, contextual information, ambiguity, and noise. Recent studies have begun exploring modality factorization to tackle this issue ?Hazarika et al. (2020); Yang et al. (2022a); Li et al. (2023b). For example, MISA Hazarika et al. (2020) decomposes modality features into modality-invariant and modality-specific spaces. FDMER Yang et al. (2022a) uses contrastive constraints to encourage semantic separation, while DMD Li et al. (2023b) introduces dynamic graphs for disentangled knowledge distillation. However, these factorization efforts still face two core limitations:

(*i*) their semantic granularity is limited to modality-level separation, ignoring finer internal components like ambiguity or irrelevant noise; and (*ii*) they lack effective mechanisms to recombine the disentangled factors for accurate sentiment reasoning, especially under modality conflict or degradation.

To address these challenges, we propose a novel Dynamic Semantic Routing Framework (DSRF) for multimodal sentiment analysis. Our framework incorporates three dedicated modules to tackle the semantic complexity, reconstruction, and inconsistency issues in multimodal emotion modeling. First, we introduce a hierarchical semantic factorization module that explicitly decomposes each modality representation into four semantically grounded components: *primary emotion*, *contextual cue*, *ambiguity*, and *noise*, enabling fine-grained understanding and targeted supervision. Second, we propose a semantic dynamic routing interaction mechanism, which models high-order interactions between these factors via a capsule-inspired routing process, thereby reconstructing expressive modality representations with adaptive composition. Third, we design an uncertainty-aware semantic fusion strategy, which estimates the reliability of each semantic factor and dynamically adjusts their cross-modal contributions, ensuring robust sentiment prediction even under modality inconsistency or noise. We validate the effectiveness of DSRF on four widely-used MSA benchmarks, where it consistently outperforms state-of-the-art baselines in terms of both accuracy and robustness.

## 2 RELATED WORK

### 2.1 MULTIMODAL SENTIMENT ANALYSIS

Multimodal Sentiment Analysis (MSA) focuses on inferring human emotional states by combining cues from various sources such as text, speech, and visual expressions. Compared to unimodal sentiment classification, MSA presents additional challenges due to the necessity of effectively aligning, interpreting, and integrating heterogeneous modalities with different temporal and semantic characteristics. To address these challenges, prior research has extensively explored interaction modeling and feature fusion strategies Zadeh et al. (2017; 2018a); Tsai et al. (2019); Hazarika et al. (2020); Han et al. (2021); Sun et al. (2022); Li et al. (2023b). For example, CubeMLP Sun et al. (2022) performs axis-specific fusion using separate multi-layer perceptrons, enabling local feature aggregation in multiple dimensions. However, such models usually assume that all modalities are available, making them less reliable in practical deployment scenarios where missing or corrupted modalities are common. Despite these advances, existing approaches tend to treat modality semantics holistically, ignoring the internal structure of modality semantics. In contrast, our work decomposes each modality into multiple interpretable semantic components, allowing the model to selectively emphasize sentiment-relevant factors and suppress irrelevant noise. We propose a novel dynamic semantic routing framework, which includes three dedicated modules: a hierarchical semantic factorization module that disentangles modality representations into four interpretable factors — primary emotion, contextual cue, ambiguity, and noise; (2) a semantic factor dynamic routing Reconstruction Mechanism that composes semantic subspaces through a capsule-style routing scheme to reconstruct refined modality features; and (3) an uncertainty-aware semantic fusion strategy that adaptively weighs semantic contributions across modalities based on their estimated reliability.

### 2.2 FACTORIZED REPRESENTATION LEARNING

Factorized representation learning aims to disentangle learned features into semantically meaningful components that reflect different underlying generative factors. This decomposition enhances the interpretability and flexibility of the learned embeddings, and has been widely adopted in generative models such as variational autoencoders (VAEs) Bousmalis et al. (2016) and conditional GANs Odena et al. (2017). FactorVAE Kim & Mnih (2018), for example, enforces factorial independence to achieve dimension-wise separation in the latent space. In the MSA domain, factorization has recently gained attention as a means to improve generalization and robustness. FDMER Yang et al. (2022a) introduces inter-modality consistency losses to encourage modality-invariant and private feature learning. DMD Li et al. (2023b) leverages dynamic graphs and knowledge distillation to align disentangled representations across modalities, while MFSA Yang et al. (2022c) separates modality-specific and shared features to capture complementary affective information. Nevertheless, most existing factorization approaches still suffer from two key limitations: (i) their supervision signals are often weak or indirect, making it difficult to enforce functional specialization in each factor;

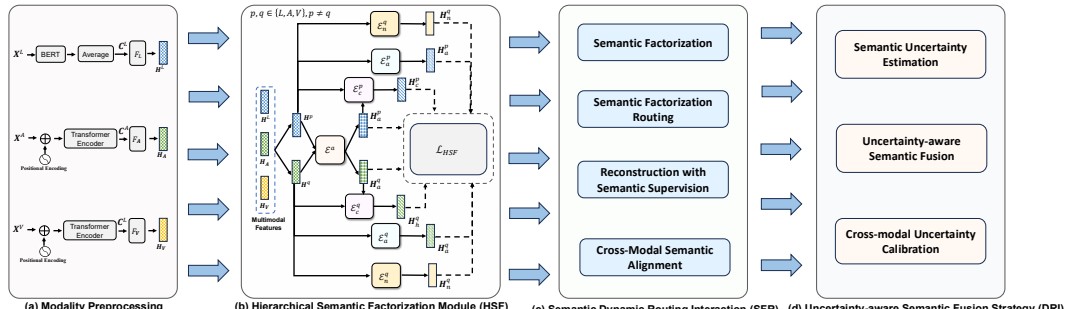

Figure 1: The overview of the proposed **Dynamic Semantic Routing Framework (DSRF)** for multimodal sentiment analysis. Each modality is first disentangled into four semantic factors via the *Hierarchical Semantic Factorization Module*, producing interpretable sub-representations: primary emotion, contextual cue, ambiguity, and noise. These factors are then dynamically routed and aggregated by the *Semantic Factor Dynamic Routing Reconstruction Mechanism* to reconstruct enhanced modality features. Finally, the *Uncertainty-aware Semantic Fusion Strategy* estimates the reliability of each semantic factor across modalities and adaptively integrates them to produce robust sentiment predictions.

and (ii) their focus remains at the modality-level separation, overlooking the fine-grained semantic distinctions that exist within a single modality, such as ambiguity or contextual dependence. To overcome these issues, we explicitly factorizes each modality into four sentiment-relevant subspaces, and leverages multiple learning signals to regulate their semantics. Moreover, the disentangled factors are recomposed through a dynamic routing mechanism that learns hierarchical interactions, and a semantic uncertainty estimator further calibrates their fusion. This comprehensive framework enhances the model's ability to perform robust, interpretable, and context-aware sentiment analysis.

# 3 PROPOSED METHOD

## 3.1 PROBLEM FORMULATION

For a video segment that contains three distinct signals, denoted as $S = \{X^m, X^m, X^m\}$, where $X^L \in \mathbb{R}^{T_L \times d_L}$, $X^A \in \mathbb{R}^{T_A \times d_A}$, and $X^V \in \mathbb{R}^{T_V \times d_V}$ denote language, audio, and visual signals, respectively. $T_{(\cdot)}$ is the sequence length and $d_{(\cdot)}$ is the embedding dimension. We aim to recognize the utterance-level sentiments via multimodal data.

## 3.2 OVERALL FRAMEWORK

The proposed Dynamic Semantic Routing Framework (DSRF) aims to enhance multimodal sentiment understanding by modeling and composing semantic factors in a structured manner. Given multimodal inputs from language, acoustic, and visual modalities, the framework operates in three stages: **(i)** In the *Hierarchical Semantic Factorization Module*, each modality representation is disentangled into four distinct semantic components: primary emotion, contextual cue, ambiguity, and noise. This decomposition enables explicit modeling of sentiment-relevant and irrelevant information within each modality. **(ii)** These semantic factors are then passed into the *Semantic Factor Dynamic Routing Reconstruction Mechanism*, which employs a capsule-inspired routing process to selectively aggregate and reconstruct high-quality modality representations by learning dynamic inter-factor interactions. **(iii)** Finally, the reconstructed modality features are fused via the *Uncertainty-aware Semantic Fusion Strategy*, which estimates the reliability of each semantic factor and adaptively integrates them across modalities for robust sentiment prediction. This design allows the model to adapt to modality inconsistency, enhance semantic interpretability, and maintain performance under ambiguity or partial information scenarios.

### 3.3 MODALITY PREPROCESSING

The modality $\boldsymbol{X}^m$ with $m \in \Phi$, is processed through a 1D temporal convolutional layer with kernel size $3 \times 3$. Then, positional embeddings Vaswani et al. (2017) are added to generate the preliminary representations, which are expressed as $\boldsymbol{C}^m = \boldsymbol{W}_{3\times3}(\boldsymbol{X}^m) + PE(T^m, d) \in \mathbb{R}^{T^m \times d}$. The $\boldsymbol{C}^m$ is fed into a Transformer Vaswani et al. (2017) encoder $\mathcal{F}^m(\cdot)$, and the last element of its output is denoted as representation $\boldsymbol{H}^m = \mathcal{F}^m(\boldsymbol{C}^m) \in \mathbb{R}^d$.

### 3.4 HIERARCHICAL SEMANTIC FACTORIZATION MODULE

Multimodal sentiment analysis often requires models to capture heterogeneous emotional expressions from language, vision, and acoustic modalities. However, most existing models encode each modality into a single latent vector Tsai et al. (2019); Yu et al. (2021), which implicitly entangles multiple types of semantic information such as emotion, context, ambiguity, and noise. While some prior works have attempted representation disentanglement, such as using modality-specific gates Hazarika et al. (2020); Li et al. (2023b) or disentangling multimodal interactions Rahman et al. (2020), they often focus on inter-modality fusion or sentiment-invariant factors, and lack fine-grained control over the internal structure of each modality. To address these limitations, we introduce the Hierarchical Semantic Factorization (HSF) module, which explicitly disentangles each modality's representation into four functionally meaningful and interpretable factors: primary emotion, contextual cue, ambiguity, and noise. This hierarchical design enables the model to better structure and utilize heterogeneous semantic cues, enhancing its capability to handle ambiguous or noisy multimodal input in the presence of modality uncertainty.

Specifically, we decompose each modality into four distinct types of representations: (i) *emotion representation* captures the core emotional intention conveyed by the modality, such as anger, happiness, or sadness, and typically carries the most discriminative signal for classification. (ii) tex-titcontextual representation reflects background or auxiliary information that does ous contexts. (iii) *ambiguity representation* models sources of semantic confusion, such as irony, polysemy, or conflicting cross-modal signals, which contribute to uncertainty in sentiment judgment. (iv) *noise representation* encodes task-irrelevant or harmful signals that may arise from background clutter, modality-specific artifacts, or other non-sentiment features. For any modality $m \in \Phi$, we obtain the emotion representation representation by a modality-shared encoder, denoted as $\boldsymbol{R}_e^m = \mathcal{E}_e(\boldsymbol{H^m})$, a contextual representation by a modality-specific encoder, denoted as $\boldsymbol{r}_c^m = \mathcal{E}_c^m(\boldsymbol{H}^m, \boldsymbol{r}_{ho}^m)$, and a ambiguity representation by a modality-specific encoder, denoted as $\boldsymbol{r}_a^m = \mathcal{E}_a^m(\boldsymbol{H}^m)$, and a noise representation by a modality-specific encoder, denoted as $\boldsymbol{r}_n^m = \mathcal{E}_n^m(\boldsymbol{H}^m)$ All encoders are composed of multi-layer perceptrons with the ReLU activation. To ensure functional and semantic separation among these factors, We designed the following constraint paradigm.

**Primary emotion factor.** The factor $z_e$ is responsible for encoding the dominant sentiment-related signals. To ensure its discriminative power, we impose direct supervision by attaching a sentiment classifier and optimizing the cross-entropy loss:

$$\mathcal{L}_{emo} = -\sum_y \hat{y} \log p(y|z_e). \qquad (1)$$

In addition, we maximize the mutual information between $z_e$ and the ground-truth labels, which encourages $z_e$ to retain sentiment-relevant information and prevents it from being contaminated by auxiliary or noisy cues.

**Contextual cue factor.** The factor $z_c$ is expected to capture auxiliary information such as background, discourse, or situational context. To explicitly encourage contextual reconstruction, we employ an auxiliary decoder that reconstructs local contextual signals from $z_c$:

$$\mathcal{L}_{ctx} = \mathbb{E}\left[\|x_{context} - \hat{x}_{context}(z_c)\|^2\right]. \qquad (2)$$

Moreover, to avoid redundancy with the primary emotion factor, we impose an orthogonality constraint by minimizing the mutual information between $z_c$ and $z_e$:

$$I(z_c; z_e) \approx 0. \qquad (3)$$

This ensures that contextual cues complement, rather than overlap with, sentiment signals.

**Ambiguity factor.** The ambiguity factor $z_a$ is designed to represent uncertain or conflicting evidence across modalities. To explicitly separate ambiguity from dominant sentiment, we adopt a margin-based contrastive objective:

$$\mathcal{L}_{amb} = \max(0, \gamma - d(z_a, z_e)), \tag{4}$$

where $d(\cdot)$ denotes a distance metric and $\gamma$ is a margin hyperparameter. Furthermore, to preserve its inherently uncertain nature, we regularize $z_a$ with an entropy maximization term, keeping its distribution broad and preventing premature collapse:

$$H(z_a) = -\sum p(z_a) \log p(z_a). \tag{5}$$

**Noise factor.** The factor $z_n$ is intended to absorb modality-specific noise or irrelevant details, ensuring that other factors remain clean and informative. each modality has its own unique noise representation while remaining consistent across samples of the same modality. Therefore, the noise representation is constrained as follows:

$$
\begin{aligned}
\mathcal{L}_{CL}^n = &-\sum_{i=1}^{N} \sum_{m \in \Phi} \log \frac{\exp\left(\text{sim}(\boldsymbol{R}_{i,m}^n, \boldsymbol{R}_{i,\bar{m}}^n)/\tau\right)}{\sum_{j=1}^{N} \sum_{b \in \Phi} \exp\left(\text{sim}(\boldsymbol{R}_{i,m}^n, \boldsymbol{R}_{j,b}^n)/\tau\right)} \\
&- \sum_{i=1}^{N} \sum_{a \in \Phi} \sum_{b \in \Phi, b \neq a} \log \frac{\exp\left(\text{sim}(\boldsymbol{R}_{i,a}^n, \boldsymbol{R}_{i,b}^n)/\tau\right)}{\sum_{j=1}^{N} \sum_{c \in \Phi} \exp\left(\text{sim}(\boldsymbol{R}_{i,a}^n, \boldsymbol{R}_{j,c}^n)/\tau\right)},
\end{aligned}
\tag{6}
$$

where $\text{sim}(\cdot, \cdot)$ denotes the similarity function (e.g., cosine similarity) and $\tau$ is a temperature parameter controlling distribution sharpness.

**Training objective** The final training objective of the HSF integrates the above factor-specific losses with an independence regularizer that enforces disentanglement among all latent variables:

$$\mathcal{L}_{HSF} = \mathcal{L}_{emo} + \alpha \mathcal{L}_{ctx} + \beta \mathcal{L}_{amb} + \gamma \mathcal{L}_n. \tag{7}$$

## 3.5 SEMANTIC DYNAMIC ROUTING INTERACTION

While semantic factorization improves representation interpretability, the challenge remains in effectively integrating these heterogeneous factors to form expressive, high-order multimodal representations. Existing reconstruction strategies Pham et al. (2019); Lian et al. (2023) either rely on fixed fusion operators (*e.g.*, concatenation, averaging) or perform modality translation in a cyclic manner, which may fail to capture complex interactions across semantic components, especially under dynamic or uncertain conditions. Additionally, these methods typically adopt static or linear combination schemes, ignoring the fact that the relative importance of each semantic factor may vary across samples and modalities. For example, when the visual signal is noisy, the model should route more weight through the primary emotion and contextual cues from the language modality while down-weighting the visual ambiguity factor. Static fusion fails to capture this behavior. To address this challenge, we propose a semantic Dynamic Routing Interaction (DRI) mechanism. Inspired by capsule networks Sabour et al. (2017), SFR treats each semantic factor as a low-level capsule and dynamically routes them to high-level latent representations through an iterative agreement mechanism. This design enables the model to capture complex, sample-specific interactions among factors and encourages the emergence of compositional sentiment structures. In particular, SFDR is robust to missing or unreliable factors, as routing weights are updated adaptively based on their contribution strength.

Specifically, each representation $r_i^m$ with $i \in \{e, c, a, n\}$ is projected to a capsule input space $v_i = W_i^{caps} r_i^m \in \mathbb{R}^{d'}$, and routed into $K$ output capsules $\{u_k\}_{k=1}^{K}$ via an iterative process. At the start, we initialize coupling logits $b_{ik} = 0$ for all factor-capsule pairs. At each iteration, we compute the normalized coupling weights using softmax:

$$c_{ik} = \frac{\exp(b_{ik})}{\sum_j \exp(b_{ij})}. \tag{8}$$

Next, the output capsule $u_k$ is computed by a non-linear squashing function over the weighted sum of inputs:

$$s_k = \sum_i c_{ik} v_i, \quad u_k = \frac{\|s_k\|^2}{1 + \|s_k\|^2} \cdot \frac{s_k}{\|s_k\|}. \tag{9}$$

The routing logits are then updated using the agreement between the input and output:

$$b_{ik} \leftarrow b_{ik} + v_i^\top u_k. \tag{10}$$

After $T$ iterations, we concatenate the $K$ output capsules and apply a reconstruction network:

$$\hat{h}^m = \text{MLP}(u_1 \| u_2 \| \dots \| u_K). \tag{11}$$

To ensure that the reconstructed representation retains semantic fidelity, we introduce a reconstruction loss:

$$\mathcal{L}_{DRI} = \|\hat{h}^m - h^m\|_2^2. \tag{12}$$

### 3.6 UNCERTAINTY-AWARE SEMANTIC FUSION STRATEGY

Despite the progress in modular representation and routing, the final sentiment prediction still requires an effective semantic fusion strategy across modalities. Traditional multimodal fusion methods typically assume equal modality contributions or compute attention scores based solely on content similarity Hazarika et al. (2020); Tsai et al. (2019). However, such methods do not account for the reliability or confidence of each semantic factor, especially under noisy or missing modality conditions. For instance, in cases where the audio modality is absent or severely corrupted, assigning it a non-trivial fusion weight may introduce noise into the decision boundary. Similarly, even within the same modality, ambiguity or noise factors should not be treated equally as primary emotion or context. Prior fusion methods ignore these uncertainties, leading to overfitting to unreliable factors or under-utilization of valuable semantics. To mitigate this issue, we introduce the Uncertainty-aware Semantic Fusion (USF) strategy. It estimates a confidence score for each semantic factor across modalities based on its internal consistency and informativeness. These scores are used to reweight factors before fusion adaptively. By incorporating entropy-based regularization, the model avoids overconfident predictions and encourages diversity in fusion behavior. This uncertainty-aware design enables the model to make robust decisions even when some modalities or factors are unreliable or missing.

**Semantic Uncertainty Estimation.**  To quantify the reliability of each semantic factor, we define an uncertainty vector $u^m = [u_e^m,\ u_c^m,\ u_a^m,\ u_n^m]$. Specifically, $u_e^m$ measures the sensitivity of $r_e^m$ to ambiguous signals, computed via the KL divergence between predictions before and after perturbing $r_a^m$; $u_c^m$ reflects contextual inconsistency, estimated by comparing predictions from $r_e^m$ and $r_e^m + r_c^m$; $u_a^m$ is defined as the entropy of predictions from $r_a^m$, representing semantic fuzziness; and $u_n^m$ is the norm of $r_n^m$, serving as a proxy for noise intensity. These uncertainties are transformed into attention weights by $\alpha^m = \text{softmax}(-u^m)$, so that more uncertain components receive lower weights during fusion.

**Uncertainty-aware Semantic Fusion.**  The semantic-aware fused representation of modality $m$ is obtained by reweighting the subspaces:

$$\hat{r}^m = \sum_{i \in \{e,c,a,n\}} \alpha_i^m \cdot r_i^m \tag{13}$$

Additionally, to enhance stability and interpretability, we introduce a gated residual fusion mechanism centered on the primary emotion space:

$$\hat{r}^m = r_e^m + g^m \cdot (r_c^m + r_a^m), \quad g^m = \sigma(W_g[u_c^m;\ u_a^m]) \tag{14}$$

where $g^m$ weights contextual and ambiguous factors by their uncertainty.

**Cross-modal Uncertainty Calibration.**  To model inter-modal agreement on ambiguity, we refine each modality's uncertainty vector using a cross-modal calibration term, which allows the model to identify shared uncertainty patterns across modalities and reduce decision confidence in highly ambiguous cases. The above process is expressed as

$$\tilde{u}^m = u^m + \lambda \sum_{n \neq m} \text{sim}(r_a^m, r_a^n) \tag{15}$$

| Methods | MOSI Dataset | | | | | MOSEI Dataset | | | | |
|---|---|---|---|---|---|---|---|---|---|---|
| | $MAE\downarrow$ | $Corr\uparrow$ | $Acc_7\uparrow$ | $Acc_2\uparrow$ | $F1\uparrow$ | $MAE\downarrow$ | $Corr\uparrow$ | $Acc_7\uparrow$ | $Acc_2\uparrow$ | $F1\uparrow$ |
| TFN Zadeh et al. (2017)[†] | 0.947 | 0.673 | 34.46 | 77.99/79.08 | 77.95/79.11 | 0.572 | 0.714 | 51.60 | 78.50/81.89 | 78.96/81.74 |
| LMF Liu et al. (2018)[†] | 0.950 | 0.651 | 33.82 | 77.90/79.18 | 77.80/79.15 | 0.576 | 0.717 | 51.59 | 80.54/83.48 | 80.94/83.36 |
| ICCN Sun et al. (2020) | 0.862 | 0.714 | 39.0 | -/83.0 | -/83.0 | 0.565 | 0.713 | 51.6 | -/84.2 | -/84.2 |
| MulT Tsai et al. (2019)[†] | 0.879 | 0.702 | 36.91 | 79.71/80.98 | 79.63/80.95 | 0.559 | 0.733 | 52.84 | 81.15/84.63 | 81.56/84.52 |
| MISA Hazarika et al. (2020)[†] | 0.776 | 0.778 | 41.37 | 81.84/83.54 | 81.82/83.58 | 0.568 | 0.724 | - | 82.59/84.23 | 82.67/83.97 |
| Self-MM Yu et al. (2021)[†] | 0.708 | 0.796 | 46.67 | 83.44/85.46 | 83.36/85.43 | 0.531 | 0.764 | 53.87 | 83.76/85.15 | 83.82/84.90 |
| MMIM Han et al. (2021)[*] | 0.718 | 0.797 | 46.64 | 83.38/85.82 | 83.29/85.81 | 0.537 | 0.759 | 53.42 | 82.08/85.14 | 82.51/85.11 |
| MSG Lin & Hu (2023) | 0.748 | 0.782 | 47.3 | -/85.6 | /85.6 | 0.583 | 0.787 | 52.8 | -/85.4 | -/85.4 |
| DMD Li et al. (2023b)[*] | - | - | 43.9 | - /84.9 | - /85.0 | - | - | 53.1 | - /85.2 | - /85.2 |
| MIM Zeng et al. (2023)[*] | 0.718 | 0.792 | 46.4 | - /84.8 | - /84.8 | 0.579 | 0.779 | 51.8 | - /85.7 | - /85.6 |
| DTN Zeng et al. (2024)[*] | 0.716 | 0.790 | 47.5 | - /85.1 | - /85.1 | 0.572 | 0.765 | 52.3 | - /85.5 | - /85.5 |
| **DSRF (Ours)** | **0.679** | **0.805** | **49.72** | **84.02/86.21** | **84.25/86.50** | **0.512** | **0.790** | **54.87** | **84.91/86.12** | **84.95/86.29** |

Table 1: Comparison results on the MOSI and MOSEI datasets. †: the results from Mao et al. (2022); ∗: the results are reproduced from the open-source codebase with hyper-parameters provided in original papers. For $Acc_2$ and $F1$, we have two sets of non-negative/negative (left) and positive/negative (right) evaluation results. Bold represents the best results, respectively.

| Methods | SIMS Dataset | | | | | SIMSv2 Dataset | | | | |
|---|---|---|---|---|---|---|---|---|---|---|
| | $MAE\downarrow$ | $Corr\uparrow$ | $Acc_2\uparrow$ | $Acc_3\uparrow$ | $Acc_5\uparrow$ | $F1\uparrow$ | $MAE\downarrow$ | $Corr\uparrow$ | $Acc_2\uparrow$ | $Acc_3\uparrow$ | $Acc_5\uparrow$ | $F1\uparrow$ |
| TFN Zadeh et al. (2017)[†] | 0.432 | 0.591 | 78.3 | 65.12 | 39.3 | 78.62 | 0.329 | 0.640 | 77.95 | 70.21 | 51.93 | 77.74 |
| LMF Liu et al. (2018)[†] | 0.441 | 0.576 | 77.77 | 64.68 | 40.53 | 77.88 | 0.367 | 0.557 | 74.18 | 64.90 | 47.79 | 73.88 |
| MulT Tsai et al. (2019)[*] | 0.453 | 0.564 | 78.56 | 64.77 | 37.94 | 79.66 | 0.304 | 0.705 | 79.3 | 72.63 | 53.29 | 79.43 |
| Self-MM Yu et al. (2021)[†] | 0.425 | 0.595 | 80.04 | 65.47 | 41.53 | 80.44 | 0.322 | 0.678 | 79.11 | 72.34 | 53.0 | 79.05 |
| CENet Wang et al. (2022)[†] | 0.471 | 0.534 | 77.90 | 62.58 | 33.92 | 77.53 | 0.310 | 0.699 | 79.56 | 73.10 | 53.04 | **79.63** |
| ALMT Zhang et al. (2023)[*] | 0.408 | 0.594 | 78.77 | 65.86 | 43.11 | 78.71 | 0.308 | 0.700 | 79.59 | 71.86 | 52.90 | 79.51 |
| DMD Li et al. (2023b)[*] | 0.412 | 0.586 | 78.33 | 65.23 | 44.26 | 79.21 | 0.305 | 0.702 | 78.87 | 72.01 | 53.18 | 79.21 |
| MIM Zeng et al. (2023)[*] | 0.420 | 0.592 | 78.98 | 65.12 | 44.98 | 78.70 | 0.310 | 0.694 | 77.56 | 71.45 | 52.87 | 78.56 |
| DTN Zeng et al. (2024)[*] | 0.419 | 0.593 | 79.45 | 65.67 | 44.26 | 79.47 | 0.302 | 0.701 | 78.29 | 72.56 | 53.71 | 78.12 |
| **DSRF (Ours)** | **0.397** | **0.610** | **81.03** | **67.11** | **45.86** | **81.23** | **0.291** | **0.721** | **80.61** | **74.34** | **55.39** | 80.04 |

Table 2: Comparison results on the SIMS and SIMSv2 datasets. †: the results from Mao et al. (2022); ∗: the results are reproduced from the open-source codebase with hyper-parameters provided in original papers. Bold represents the best results, respectively.

**Final Fusion and Prediction.** All modality-specific fused vectors $\hat{r}^m$ are aggregated through a multimodal fusion module (e.g., concatenation, attention, or Transformer):

$$\hat{r} = \text{Fuse}(\hat{r}^v,\ \hat{r}^a,\ \hat{r}^t), \quad \hat{y} = \text{Softmax}(W \cdot \hat{r}) \tag{16}$$

The USF module contributes to the overall training objective through an uncertainty-aware regularization term:

$$\mathcal{L}_{\text{USF}} = \text{KL}\left(p(y|\hat{r}^m) \parallel p(y|r_e^m)\right) + \gamma \cdot \sum_m \mathcal{H}(\hat{y}^m) \tag{17}$$

Finally, the overall training objective Ltotal is expressed as $\mathcal{L}_{total} = \mathcal{L}_{task} + \mathcal{L}_{HSF} + \mathcal{L}_{SFR} + \mathcal{L}_{DRI}$, where $\mathcal{L}_{task}$ is the standard cross-entropy loss.

## 4 EXPERIMENT

### 4.1 BENCHMARKS AND EVALUATION METRICS

We conduct experiments on four publicly benchmark datasets of MSA, including MOSI Zadeh et al. (2016), MOSEI Zadeh et al. (2018b), SIMS Yu et al. (2020), and SIMSv2 Liu et al. (2022b). In

contrast to the MOSI and MOSEI, both SIMS and SIMSv2 prioritize balanced modality-specific sentiment dominance, avoiding a clear trend where any single modality dominates emotional expression. This design better validates the effectiveness of MODS. We use the accuracy of 3-class ($Acc_3$) and 5-class ($Acc_5$) on SIMS and SIMSv2, the accuracy of 7-class ($Acc_7$) on MOSI and MOSEI, and the accuracy of 2-class ($Acc_2$), Mean Absolute Error ($MAE$), Pearson Correlation ($Corr$), and F1-score ($F1$) on all datasets. In particular, higher values indicate better performance for all metrics except $MAE$.

### 4.2 FEATURE EXTRACTION AND IMPLEMENTATION DETAILS

For the textual modality, we adopt a pre-trained BERT encoder Devlin (2018) to obtain contextualized embeddings, where each input utterance is mapped to a 768-dimensional hidden representation. Regarding the visual modality, facial expression features are extracted using the Facet toolkit Baltrušaitis et al. (2016), producing 35-dimensional vectors that encode facial action units associated with emotion-related muscle movements. For the acoustic modality, we utilize COVAREP Degottex et al. (2014) to extract 74-dimensional low-level descriptors, capturing prosodic and vocal tract characteristics. All experiments are conducted using the PyTorch framework, and models are trained on four NVIDIA Tesla V100 GPUs. For MOSI, MOSEI, SIMS, and SIMSv2, the detailed hyper-parameter settings are as follows: the learning rates are $\{3e-5, 1e-5, 1e-5, 1e-5\}$, the batch sizes are $\{32, 64, 32, 32\}$, the hidden size are $\{128, 128, 64, 128\}$, and weight decay are $\{1e-3, 1e-3, 1e-2, 1e-2\}$. The hyper-parameters are determined based on the validation set.

### 4.3 COMPARISON WITH STATE-OF-THE-ART METHODS

To demonstrate the superior performance of DSRF, we compare it with some representative SOTAs, which are LMF Liu et al. (2018), MFN Tsai et al. (2018), ICCN Sun et al. (2020), MulT Tsai et al. (2019), MISA Hazarika et al. (2020), FDMER Yang et al. (2022a), DMD Li et al. (2023b), MIM Zeng et al. (2023), and DTN Zeng et al. (2024).

As shown in Tables 1 and 2, we summarize several key observations from the comparative results. (i) Our proposed method achieves consistent and significant improvements over existing baselines across most evaluation metrics on all benchmark datasets. This performance gain primarily stems from the structured semantic modeling introduced by our framework. Specifically, the hierarchical factorization mechanism enables the extraction of four distinct and functionally meaningful representations—primary emotion, contextual cue, ambiguity, and noise—thereby allowing the model to better isolate sentiment-relevant signals while mitigating irrelevant or misleading features. In contrast to traditional holistic encoders, our approach explicitly disentangles the sentiment structure of each modality, leading to more informative and interpretable representations. Additionally, by incorporating dynamic routing and uncertainty estimation into reconstruction and fusion stages, the framework ensures that these semantic components are utilized in an adaptive and robust manner. (ii) Second, in comparison to MIM Zeng et al. (2023), which utilizes unimodal purification and cross-modal feature gating to filter out noisy signals and amplify useful ones, our method demonstrates a more principled and fine-grained treatment of modality internals. While MIM applies coarse control over modality-level information flow, it does not explicitly model the internal composition of sentiment cues. Our hierarchical semantic factorization module, in contrast, allows each modality to be decomposed into interpretable subspaces, capturing latent sentiment structures in greater depth. This leads to a more flexible and accurate representation, especially in scenarios with conflicting or incomplete modality cues. (iii) Third, compared with DTN Zeng et al. (2024), which focuses on aligning modality representations through distribution-level disentanglement and consistency learning, our approach offers more comprehensive modeling in both representation and fusion stages. Beyond enforcing distributional alignment, we introduce a dynamic routing reconstruction mechanism inspired by capsule networks, enabling higher-order composition of semantic factors through learned routing paths. Moreover, our uncertainty-aware semantic fusion strategy further enhances robustness by estimating the reliability of each factor and adaptively integrating them across modalities. This dual-level optimization—semantic disentanglement and reliability-aware fusion—leads to stronger generalization and more accurate sentiment predictions, particularly under noisy or ambiguous input conditions.

Table 3: Ablation results on the MOSI dataset.

| Models | Metrics | | | | |
|---|---|---|---|---|---|
| | MAE ↓ | Corr ↑ | Acc-7 ↑ | Acc-2 ↑ | F1 ↑ |
| **DSRF (Full)** | **0.679** | **0.805** | **49.72** | **84.02/86.21** | **84.25/86.50** |
| Importance of Modalities | | | | | |
| w/o language | 1.226 | 0.312 | 24.5 | 52.6/54.5 | 52.7/54.7 |
| w/o Audio | 0.748 | 0.786 | 44.7 | 81.4/83.6 | 81.6/83.7 |
| w/o Visual | 0.713 | 0.811 | 45.4 | 84.5/86.2 | 84.7/86.3 |
| Importance of Proposed Components | | | | | |
| w/o HSF | 0.714 | 0.772 | 46.02 | 80.2/82.4 | 80.4/82.5 |
| w/o SFR | 0.709 | 0.785 | 47.77 | 82.4/84.7 | 82.5/84.8 |
| w/o DRI | 0.712 | 0.796 | 47.39 | 82.9/85.1 | 83.0/85.3 |

## 4.4 ABLATION STUDIES

To verify the effectiveness of each core component and modality within the proposed DSRF framework, we conduct a series of detailed ablation studies, as summarized in Table 3. The analysis focuses on evaluating how each module and modality contributes to overall performance. (i) We first examine the role of each input modality by independently removing language, visual, or acoustic signals from the framework. The results show a consistent and significant performance drop in all settings, confirming that cross-modal complementarity is critical for accurate sentiment prediction. Among them, the language modality contributes the most, as it often carries explicit emotional expressions and semantically rich sentiment cues, while acoustic and visual modalities provide essential paralinguistic and contextual support. (ii) To assess the contribution of the hierarchical semantic factorization module, we eliminate it and directly use the original modality embeddings without semantic disentanglement. The performance degradation is evident, demonstrating that modeling the internal structure of each modality through four semantic factors—primary emotion, contextual cue, ambiguity, and noise—leads to more refined and task-relevant representations. Without this decomposition, the model is less capable of isolating sentiment-relevant components from modality-specific noise, leading to less reliable multimodal fusion. (iii) We then ablate the semantic factor dynamic routing reconstruction mechanism and instead perform naive concatenation or averaging over the semantic factors. The resulting performance drop highlights the importance of dynamic interaction modeling among the semantic components. The routing mechanism adaptively learns which factors should be emphasized or suppressed based on high-order compositionality, thereby improving the quality of reconstructed modality features. This step is especially critical in scenarios with ambiguous or conflicting semantic signals. (iv) Finally, we assess the uncertainty-aware semantic fusion strategy by replacing it with equal-weighted fusion. The results reveal a notable decline in accuracy, indicating that dynamically weighting semantic factors based on their estimated reliability is essential for robust sentiment reasoning. The fusion strategy enables the model to downweight noisy or ambiguous components and focus on trustworthy semantic cues, particularly under modality inconsistency or partial degradation.

## 5 CONCLUSION

In this paper, we introduce a Dynamic Semantic Routing Framework (DSRF) to address semantic entanglement, cue ambiguity, and modality inconsistency in multimodal sentiment analysis. Specifically, we propose a Hierarchical Semantic Factorization module that disentangles each modality into four interpretable factors—primary emotion, contextual cue, ambiguity, and noise—thus enabling more fine-grained semantic representation. Furthermore, we design a Semantic-factor Dynamic Routing reconstruction mechanism that leverages capsule-style iterative routing to capture higher-order interactions among semantic factors and reconstruct robust modality representations. Finally, we present an Uncertainty-aware Semantic Fusion strategy that estimates the reliability of each semantic factor and adaptively integrates multimodal information, ensuring robust predictions even under modality conflict, degradation, or absence. Extensive experiments and ablation studies on multiple benchmarks confirm the effectiveness and robustness of our framework.

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

## A  APPENDIX

### A.1  STATEMENT ON USE OF LARGE LANGUAGE MODELS

During the preparation of this manuscript, we employed large language models solely for language refinement and writing improvement. All research ideas, experimental designs, method implementations, and analysis of results were conceived and carried out independently by the authors. The models did not contribute to the generation of scientific content, data analysis, or formulation of conclusions.

